# Flexibility of Wireless Power Transfer Charging Station Using Dynamic Matching and Power Supply with Energy Dosing

**Nikolay Madzharov [1] and Nikolay Hinov [2,\*]** 

[1] Department of Electronics, Faculty of Electrical Engineering and Electronics, Technical University of Gabrovo, 4 H. Dimitar, 5300 Gabrovo, Bulgaria; madjarov@tugab.bg

[2] Department of Power Electronics, Faculty Electronic Engineering and Technology, Technical University of Sofia, 8, Kliment Ohridski Blvd, 1000 Sofia, Bulgaria

\* Correspondence: hinov@tu-sofia.bg; Tel.: +359-2965-2569

**Abstract:** The scientific and applied problems discussed in this paper are related to the development of a wireless charging station using an inductive power transfer (IPT) module power supply with energy dosing and dynamic matching. A computer simulation and an experimental study allowed the authors to define the ranges of the parameter variation of the equivalent load and to design the best matching so that maximum energy transfer is efficiency achieved. The proposed principle of energy control provides highly reliable and a flexible charging station even with a simplified system of automatic control and fault protection. A prototype charging station is developed and built to supply an inductive power transfer system that delivers 30–35 kW power over an air gap between transmitting and receiving parts measuring 50–200 mm and with a horizontal misalignment of ±200 mm. The results showed that the system can transfer the specified electrical power with about 82–92% efficiency and that the IPT module and its dynamic matching during charging exhibited a high degree of stability under a misaligned (x-y-z) condition and battery state of charge.

**Keywords:** inductive power transfer; dynamic matching; magnetic coupling; energy dosing; charging station

## 1. Introduction

Over the last decade, almost all leading automobile companies have made research and development contributions in the field of electric vehicles (EVs). One of the most widely discussed problems is related to the charging of EVs with electrical energy. The focus being on battery capacity and mileage without charging [1,2] and infrastructure and charging time [3,4]. Concerning the process of charging, several different techniques have been, and still are, under investigation [5–9]. It can be identified with the following categories, each of which has its advantages and disadvantages: regular cable charging, fast cable charging, inductive charging, and fast inductive charging.

Inductive charging provides higher level of operator safety and robustness of installation. However, energy transfer is limited by the distance between the transmitting and receiving parts. For instance, fast inductive charging technology, which provides quick energy transfer, necessitates a gap of about 10–20 cm and approximately constant vertical and horizontal misalignments. In addition, it can be said that, to achieve dynamic charging, it is necessary to use only a wireless system [10,11].

Inductive power transfer (IPT) systems contain movable and immovable parts (see Figure 1) and have three main degrees of freedom (x-y-z), though a yaw angle can have certain influence, depending upon the system design. Different gaps between transmitting and receiving blocks change the optimal

matching and result in relatively big magnetization current, additional reactive power of the system, and reduce the transferred power level and efficiency.

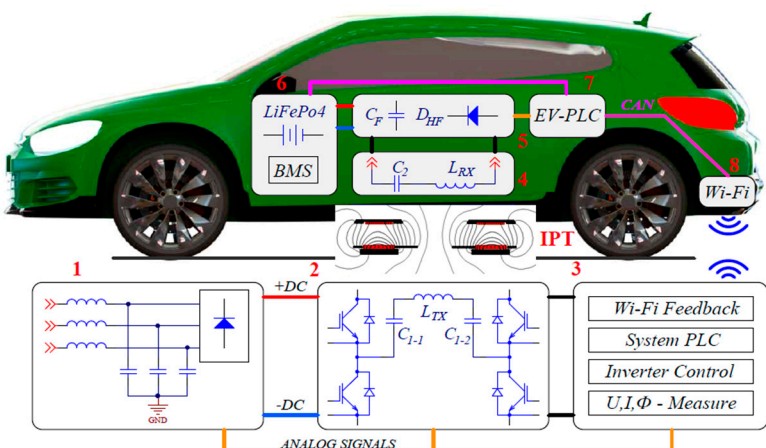

**Figure 1.** A schematic overview of inductive power transfer (IPT) charging station architecture.

Generally, two basic approaches are applied to cope with this problem. The first one is based on purely technological solutions related to the exact positioning of the receiving coil—x-y-z adjustment system, enabling transmitting and/or receiving coil's displacement, using pneumatic rear suspension, etc. [1,7,10]. This is achieved by using a mechanical system that makes the overall infrastructure more complicated.

The second method uses a change in the matching characteristics of the HF power supply. In this case matching due to careful initial setup of the system is insufficient and extra channels of adjustment are necessary. Major additional control of matching could be affected by means of frequency sweeping and/or dynamic correction of matching setup by changing of resonant capacitors value. Frequency change as a method of regulation and matching has been widely discussed in references [5,12,13]. The main conclusion is that this method is not suitable for loads with highly variable parameters, including IPT systems for EV charging, because it increases the switching losses of transistors, causes highly distorted form of current through the IPT coils, and works in inappropriate operating modes with decreased efficiency.

Another option for dynamic matching is to change the parameters of compensation elements [1,5,11,14,15]. Technically, it is appropriate to change the matching capacitor value in transmitting using electronic switches. This allows for an adjustment of the matching circuit during the charging process and in practice for the system to work with different characteristics of the battery and state of charge (SOC).

An actual trend in this area is the use of HF power supplies in which matching is done by their algorithm, i.e., without control system influence. Similar properties are displayed by the autonomous inverter with energy dosing (AI with ED). What distinguishes them from the sources we have been familiar with so far is the fact that their output power is assigned in a definite way, and, in the process of operation, it does not depend on the load parameters, thus remaining equal to the assigned value.

This paper aims to present the design and tests of fast inductive charging applications of EVs system on the basis of AI with ED. Special attention is paid to the IPT infrastructure and its dynamic and flexible matching, which transfers 35 kW through a 50–200 mm air gap and is applicable for the static and dynamic charging of EVs.

## 2. Topology of Inductive Power Transfer System

The IPT module is conducted in harmony to "SAE J2954" standard [16], and its draft is shown in Figure 2. Several parameters potentially impact the IPT module efficiency and EM field strength.

They can be conditionally separated into electricals and mechanicals, and some of their calculation is discussed in References [1,6,12,17].

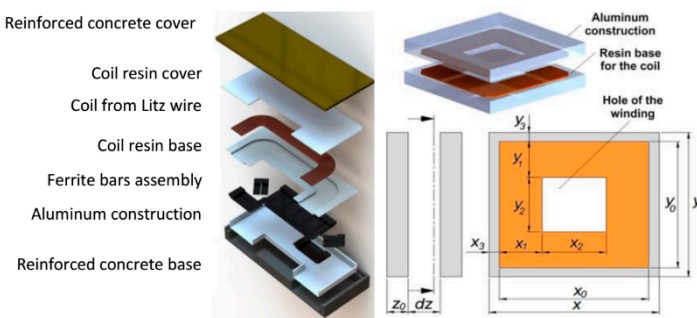

**Figure 2.** A 3D model of IPT coils.

First and foremost are the ratio of magnetic coupling *k* and mutual inductance $M_{TX\text{-}RX}$ between transmitting and receiving coils. The physical essence is related with the distribution of the magnetic flux between the coils and could be calculated by

$$M_{TX-RX} = \oint_{S_{RX}} \frac{\Phi_{TX-RX}(I_{TX})}{I_{TX}} dS_{RX} = k\sqrt{L_{TX}L_{RX}} \tag{1}$$

where $\Phi_{TX\text{-}RX}$ is the magnetic flux linkage, $S_{RX}$ is the area of the receiving side, $L_{TX}$ and $L_{RX}$ are the inductance of the transmitting and receiving coils, and $I_{TX}$ is the current through the transmitting coil.

From Equation (1), it can be seen that some of the important factors for *k* and *M* are the area $S_{RX}$ and inductances $L_{TX}$ and $L_{RX}$. The developed IPT module suggests several extreme restrictions. While the inductances can acquire values over a wide range by using a different number of turns or supplementary ferrite cores, the coupling coefficient depends mainly on the dimensions of the two coils. The magnetic coupling with rectangular coils (Figure 1) is equal to

$$k(z) \approx \left(2n/\left(4n^2 + 12z^2\right)\right)^{3/2}, \; n = (x+y)/2, \tag{2}$$

Equations (1) and (2) in Figure 3 present the dependence of *k* as a function of geometrical dimensions and the distance between the transmitting and receiving coils. The main conclusion that can be drawn is that an increase in the active width of the windings leads to an increase of *k*. There is also a strong correlation between the distance between the coils and *k*. At a distance over 200 mm, the coefficient *k* becomes smaller than 0.2, which corresponds to an ineffective energy transfer. This confirms one of the basic IPT design rules [1]—for efficient energy transfer, the distance between the coils should not exceed coil diameter divided by 4.

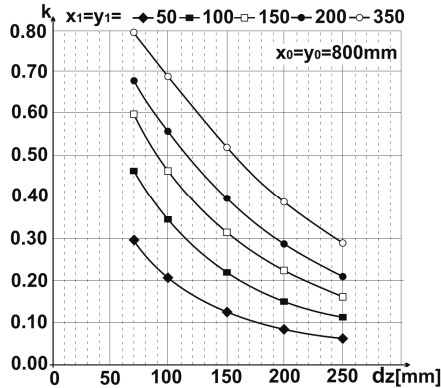

**Figure 3.** The effect of distance and IPT dimensions upon coupling coefficient (*k*).

Another key point, directly related to the efficiency of the IPT module, is coil loses—$P_{LOSS}$. The ratio of $P_{LOSS}$ toward the output power $P_{OUT}$ is equal to

$$\lambda = P_{LOSS}/P_{OUT} = 2\left(1 + \sqrt{1 + (kQ)^2}\right)/(kQ)^2 \tag{3}$$

where $Q$ is equivalent IPT quality factor.

Based on Equation (3), an analysis was made, the results of which are presented in Figure 4. It can be concluded that in order to achieve better efficiency, it is necessary that $\lambda \ll 1$. The value of losses increases dramatically at $k < 0.1$ and $Q < 10$. The implemented analysis proves that, for the reliable operation of an IPT module, it is necessary that $k > 0.2$ and $Q > 20$; in other words, $kQ = 4 \div 10$. Hence, it follows that Equation (3) could be simplified using the above values,

$$\lambda = P_{LOSS}/P_{OUT} \approx 2\left(1 + \sqrt{\beta_{IPT}^2}\right)/\beta_{IPT}^2 \approx 2/\beta_{IPT} \tag{4}$$

where $kQ = \beta_{IPT}$, $\beta_{IPT} \gg 2$.

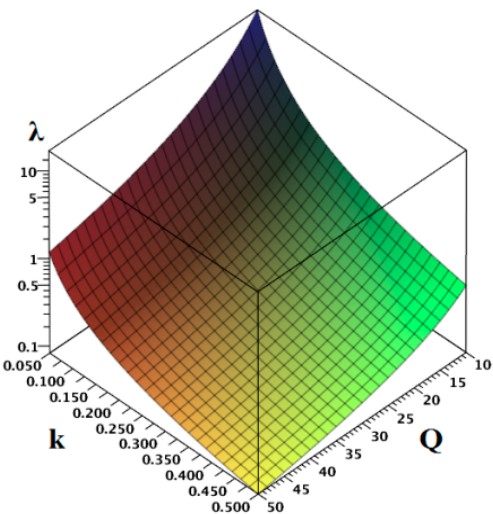

**Figure 4.** Coil losses for different $k$ and $Q$.

If the coupling coefficient $k$ has low value ($k < 0.2$), it is possible by optimizing $Q$ (increasing the inductance) to keep the ratio $kQ \gg 4$. Otherwise, the IPT module will have bad economic indicators. This condition strongly depends on the dimensions of the IPT coils ($x_0$ and $y_0$) and the distance between them $z$. If the air gap is $z < (x_0 + y_0)/4$, the transmission efficiency will become greater than 80%; if $z/[(x_0 + y_0)/2] < 0.2$, the transmission efficiency could be close to 90–95%.

## 3. Equivalent Circuits and Matching Network

To achieve maximum efficiency there should be reactive energy compensation of the transmitting and receiving parts. The IPT device can be regarded as quasi-linear quadrupole and may be presented in the form of equivalent circuits. When using a standard T-equivalent circuit, all three inductances are compensated individually, according to Figure 5.

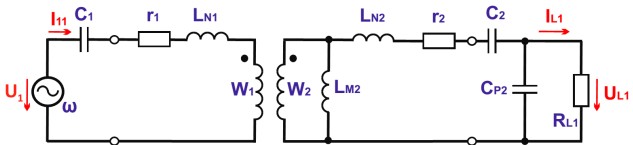

**Figure 5.** Compensation topologies of T-equivalent circuit.

Self-inductances of the two coils (when the coils are not connected to other circuits) are equal to

$$L_1 = w_1^2 g, \ L_2 = w_2^2 g, \tag{5}$$

where $g$—magnetic permeance of the coil.

When IPT is connected to the power source and the load, the elements values from Figure 5 can be calculated by the following equations:

$$L_{N1} = L_1 - Mw_1/w_2 = L_1\left(1 - k\sqrt{g_{C2}/g_{C1}}\right) \tag{6}$$

$$L_{M2} = Mw_2/w_1 = L_2 k\sqrt{g_{C1}/g_{C2}}, \tag{7}$$

$$L_{2N} = L_2 - Mw_2/w_1 = L_2\left(1 - k\sqrt{g_{C1}/g_{C2}}\right), \tag{8}$$

$$k = M/\sqrt{L_1 L_2} \tag{9}$$

$$C_1 = 1/\left[\omega^2 L_1\left(1 - k\sqrt{g_{C2}/g_{C1}}\right)\right] \tag{10}$$

$$C_2 = 1/\left[\omega^2 L_2\left(1 - k\sqrt{g_{C1}/g_{C2}}\right)\right] \tag{11}$$

$$C_{P2} = 1/\left[\omega^2 L_2 k\sqrt{g_{C1}/g_{C2}}\right] \tag{12}$$

where $g_{C1}$ and $g_{C2}$ are magnetic permeances of the transmitting and receiving coils.

Equivalent circuits with two inductances allow us to compensate the IPT with a minimum number of capacitors. In accordance with the connecting points of the compensating capacitor at the secondary side and the load, there are two possibilities: series or parallel matching (Figure 6).

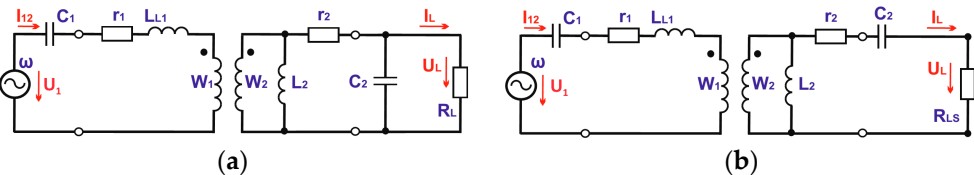

(**a**)           (**b**)

**Figure 6.** Matching of the IPT equivalent circuit with two inductances through serial capacitor at the primary side capacitor at the secondary side: (**a**) parallel and (**b**) series.

The following equations are valid for the elements of both equivalent circuits in Figure 6:

$$L_{L1} = L_2\left(1 - k^2\right) \tag{13}$$

$$n = U_1/U_L \approx I_L/I_{12} \approx w_1^*/w_2^* = k(w_1/w_2)\sqrt{g_{C1}/g_{C2}} \tag{14}$$

i.e., $n = k$ when $w_1 = w_2$

$$C_1 = 1/\omega^2 L_2\left(1 - k^2\right) = 1/\omega^2 L_{L1} \tag{15}$$

Considering $(r_2/\omega_0 L_2)^2 \ll 1$, $r_2/R_L \ll 1$ and
$C_2 = \left[(1 + r_2/R_L)^2/\omega_0^2 L_2\right]/\left[1/2 + \sqrt{1/4 - \left[(r_2 + r_2^2/R_L)/\omega_0 L_2\right]^2}\right]$ then:

$$C_2 \approx (1 + r_2/R_L)^2/\left(\omega^2 L_2\right) = 1/\omega^2 L_2 \tag{16}$$

It is reasonable to assume that, at a certain frequency (the frequency for which IPT is designed) of supply voltage—$\omega_0$ and $Z$ value, which corresponds to the combination of parameters $k_0$, $L_{10}$, $L_{20}$ ($k = k_0$, $L_{10} = L_1(k_0)$, $L_{20} = L_2(k_0)$), the values of matching elements are determined according to Equations (13)–(16).

With magnetic coupling $k_0$, the efficiency can be calculated by the following expression:

$$\eta_0 = \left(k_0^2 Q_1 Q_2\right) / \left(1 + \sqrt{1 + k_0^2 Q_1 Q_2}\right)^2 \tag{17}$$

where $Q_1$ and $Q_2$ are qualitative factors of IPT coils, including resistances $r_1$ and $r_2$, which define losses in the primary and secondary sides, i.e.,

$$Q_1 = \omega_0 L_{10} / r_1, \; Q_2 = \omega_0 L_{20} / r_2, \tag{18}$$

To obtain efficiency greater than $\eta_0$ it is necessary to fulfill the following condition:

$$k_0^2 Q_1 Q_2 \geq 4\eta_0 / (1 - \eta_0)^2 \tag{19}$$

With $Q_1 = Q_2 = Q$, there follows:

$$Q \geq (1 - \eta_0)^2 / [k_O(1 - \eta_0)] \tag{20}$$

This equation provides information about the minimum $Q$ value and, consequently, the maximum values of $r_1$ and $r_2$, so that the current efficiency be greater than $\eta_0$.

According to Equations (19) and (20), it could be proved that there is maximum value of the efficiency—$\eta_{0MAX}$, only when $R_L > \omega_0 L_{20}$, i.e., at optimum load quality factor—$Q_{L\,OPT}$.

$$Q_{L\,OPT} \approx \left(\sqrt{Q_2/Q_1}\right)/k_0 \text{ at } Q_1 = Q_2 \quad Q_{L\,OPT} \approx 1/k_0. \tag{21}$$

In the case of serial compensation, IPT is most effective for $k_0$ (respectively frequency $\omega_0$) and optimal load resistance, according to the following equation:

$$R_{LS\,OPT} = \omega_0 L_{20} / Q_{LE\,OPT} \text{ for } Q_1 = Q_2, \; R_{LS\,OPT} \approx \omega_0 L_{20} k_0. \tag{22}$$

Physical interpretation of circuits in Figures 5 and 6 is different. The analysis outcomes and the relative values of equivalent inductances are summarized in Table 1 in the following order:

- row 2—relative value of the magnetic permeances $g(k)/g(k0)$ in the range of magnetic coupling $0.2 \leq k \leq 0.6$. This relationship is identical to dependence $L_1$, $L_2 = f(k)$ at IPT idle running and also identical to $L_2 = f(k)$, at equivalent circuit with two inductances, when IPT is connected to the power supply and load (Figure 6);
- row 3—shows the relative change of mutual inductance $L_{M2}$ of T equivalent circuit (Figure 5);
- row 4—the change of leakage inductances $L_{N1}$ and $L_{N2}$ of T equivalent circuit;
- row 5—the change of leakage inductance $L_{L1}$ from equivalent circuit with two inductances. Its change is a much smaller range than the leakage inductances $L_{N1}$ and $L_{N2}$ from row 4.

**Table 1.** Relative values of the main IPT electromagnetic parameters at coupling factor $k$ in a range of 0.2–0.4.

| k | 0.2 | 0.3 | 0.4 | 0.5 | 0.6 |
|---|---|---|---|---|---|
| $g(k)/g(k_0)$ | 0.94 | 0.96 | 1 | 1.028 | 1.1 |
| $k\,g(k)/(k_0\,g(k_0))$ | 0.46 | 0.71 | 1 | 1.3 | 1.6 |
| $g(k)(1-k)/(g(k_0)(1-k_0))$ | 1.25 | 1.11 | 0.9 | 0.85 | 0.72 |
| $g(k)(1-k^2)/(g(k_0)(1-k_0^2))$ | 1.07 | 1.04 | 0.96 | 0.91 | 0.81 |

Analyzing the results recorded in Table 1, the expression of equivalent circuits with two and three inductances and their compensating circuits can lead to the following conclusions:

- there is a strong change of magnetic coupling and equivalent inductance values from current position of the coils. This requires the use of additional measures for dynamic matching at different x-y-z coils positions;
- in case of magnetic coupling change, T equivalent circuit inductances change over a wider range than in the equivalent circuit with two inductances (rows 3 and 5, Table 1);
- at *k* values in the range $0.2 < k < 0.6$, relative change of $\Delta L_{N1}(k)/\Delta L_{L1}(k)$ is about 1.68.

## 4. Power Electronics Converter and Power Control

The developed and tested inductive charging infrastructure contains four successively placed primary coils. Only one transmitting coil is used in static charging tests and four coils in dynamic tests. The power electronics circuit contains full bridge AI with ED and four electronic switches—Figure 7. Each electronic switch has only one transistor ($VT_{11}$-$VT_{13}$; $VT_{21}$-$VT_{23}$; $VT_{31}$-$VT_{33}$; $VT_{41}$-$VT_{43}$) and one series capacitor ($C_{11}$-$C_{13}$; $C_{21}$-$C_{23}$; $C_{31}$-$C_{33}$; $C_{41}$-$C_{43}$). This capacitor is part of the matching circuit of the transmitting coil. Every capacitor from the electronic switches is charged up to $U_{MAX}$ by reverse diode of the transistor, when the transistor is switched off. Later, when the transistor is switched on, a corresponding capacitor will be added to the AC circuit by the transistor and its reverse diode [8].

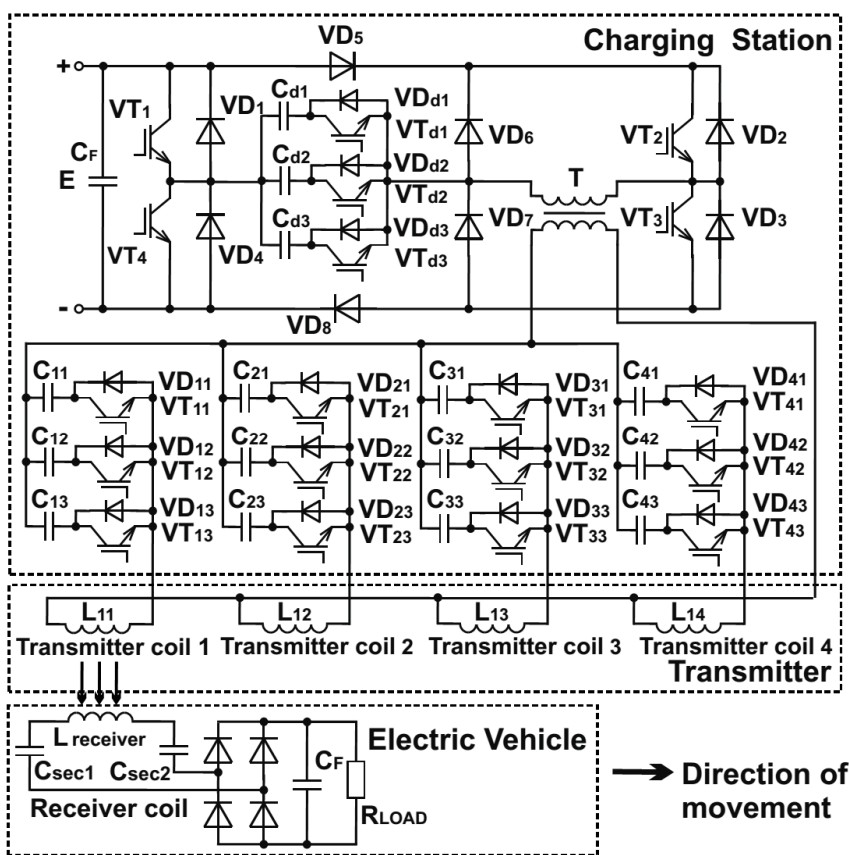

**Figure 7.** Proposed autonomous inverter with energy dosing (AI with ED) and dynamic matching circuit.

When changing any of the IPT parameters (depending on the current coil's position) or battery (in charge), the optimal matching and consequently the value of the output power and efficiency are disrupting. An essential point here is the type of matching circuit and output characteristics of the used AI with ED. For most of the ideas, schemes, design, and mathematical interpretation of electromagnetic processes, there are different author claims.

Figure 8 presents the time charts characterizing the operation of the AI with ED. There are two intervals —$(0 \div \varphi_d)$ of energy consumption from the DC power supply through capacitors $C_{d1-3}$, and $(\varphi \div \pi_d-t_0)$ of the short-circuit of the AC circuit over one of the supply lines (positive or negative).

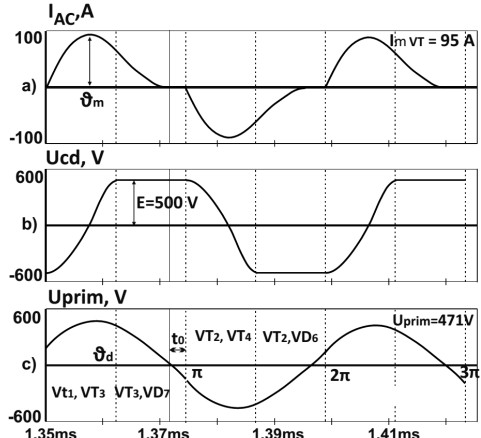

**Figure 8.** Basic time charts of AI with ED at 35 kW output power and battery as a load: the current in inverter AC diagonal; the voltage of the dosing capacitor Cd; the voltage of the transmitting coil.

Maintenance of constant $\varphi$ power in the inverter is due to the fact that the energy of the power supply is always consumed through the dosing capacitor. The voltage over the capacitor $C_d$ is fixed up to the value of the DC supply voltage ($E$) through diodes $V_{D6}$ and $V_{D7}$. Then, the power P, transmitted by the power supply, through IPT to the EV, is equal to

$$P = 4E^2 f C_d = E I_{in} = U_{out} I_{out} = const \tag{23}$$

i.e., when the operating frequency $f$, the supply voltage $E$ and the value of the dosing capacitor are constant, the power given to the load does not depend on its parameters. This internal automatism makes AI with ED very flexible as a source of HF energy in charging stations and is a prerequisite for the successful solution of many problems in matching its output parameters with the battery.

Basic analytical dependences of AI with ED have been derived on the grounds of the relationships from the harmonic analysis and the postulates of the general theory of autonomous inverters [18]. The design methods with parallel and series tank circuit have the following sequence:

1. The pause between control pulses $t_0$ is assigned, e.g., $t_0 = 0.1\pi$ (Figure 8).
2. Ratio $\omega_{po}/\omega = 1.2 - 1.4$ value is determined ($\omega_{po}$ is the natural frequency of the AC equivalent circuit, and $\omega$ is the control frequency).
3. Phase angle $\delta$ between AC output voltage and current is determined by

$$\delta > (1.2 \div 1.4)\omega_{po}/\omega.$$

4. Calculation of the quality factor of the series equivalent circuit is done by

$$Q = \omega L_E / R_E = 1/2 (\omega / \omega_{po})^2 \left[ tg\delta + \sqrt{tg^2\delta - (\omega/\omega_{po})^2} \right] \tag{24}$$

5. The dosing capacitor is calculated having in mind the power and frequency assigned.

$$C_d = P/(4E^2 f) \tag{25}$$

6.  Detuning of the equivalent AC circuit is determined by the following expression:

$$\xi_0^2 = (tg\varphi + ctg\varphi)/(tg\varphi + tg\delta) \tag{26}$$

7.  The following expression is valid for moment $\vartheta_d$ of switching on the dosing diode.

$$\vartheta_D = \pi/(\omega_{po}/\omega) - (arctg2Q\omega_{po}/\omega)/(\omega_{po}/\omega) \tag{27}$$

8.  Average current consumed by the power supply source is determined using the ratio

$$I_0 = \frac{1}{\pi} \int_0^{\pi - \vartheta_D} i(\vartheta)d\vartheta = 4EfC_d \tag{28}$$

9.  Mean and maximum values of the currents across the transistors and the reverse diodes.

$$I_{mVT_{1\div4}} = \left[E/\left((\omega_{po}/\omega)\omega L_E\right)\right]e^{-\vartheta_m/2Q}sin(\omega_{po}/\omega)\vartheta_m, \; I_{0VT_{1\div4}} = 4EfC_k\left(1 + e^{-(\pi - t_0)/2Q}\right)$$

$$I_{mVD_{1\div4}} = i(\vartheta_d) = \left[E/\left((\omega_{po}/\omega)\omega L_E\right)\right]e^{-\vartheta_d/2Q}sin(\omega_{po}/\omega)\vartheta_d, \; I_{0VD_{1\div4}} = 4EfC_de^{-(\pi - t_0)/2Q} \tag{29}$$

This design has facilitated the prediction and setting of the control algorithm, which determines unambiguously the operation mode of the inverter. In order to ensure zero current in the process of turning transistors on and off, the following procedure is obligatory: selection of a certain relationship of the frequencies $\omega_{po}/\omega$, and following fixed dephasing of the alternating current and the voltage of the AC circuit. In the research process, the last factor has turned into a basic requirement of the control system.

## 5. Experimental Investigations

The developed static and dynamic inductive charging stations were tested at power $P = 10\text{--}5$ kW, frequency $f = 18\text{--}25$ kHz, vertical air gap between transmitting and receiving coils 50–200 mm, and horizontal misalignment up to ±200 mm. In dynamic infrastructure four primary coils were built into a road and one for static infrastructure (see Figure 9) [9].

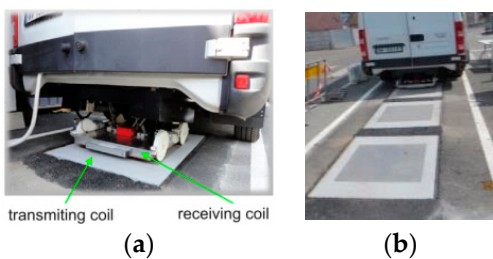

**Figure 9.** Real tests: (**a**) static and (**b**) dynamic charging of of electric vehicles (EV).

The main parts of the charging station are a full bridge AI with ED (Figure 7) and an IPT module presented in Figure 1 using the dimensions in Table 2. During the design of the primary and secondary coils, important tests were made for shielding the coils against electromagnetic field exposure. These design measures have respected the proper Directive of EC and its latest version from 2013. It is closely based on the guideline published by ICNIRP in the case of power and frequencies (ICNIRP 2010) [4,16,17].

**Table 2.** IPT dimensions (all dimensions are in mm).

| x | $x_0$ | $x_1$ | $x_2$ | $x_3$ | y | $y_0$ | $y_1$ | $y_2$ | $y_3$ | $z_0$ | z |
|---|---|---|---|---|---|---|---|---|---|---|---|
| 920 | 800 | 175 | 450 | 60 | 820 | 700 | 175 | 350 | 60 | 90 | 50–200 |

Figure 10 presents a characteristic, taken experimentally, of the HF power supply using the principle of energy dosing. This characteristic demonstrates the dependence of the output voltage and the power on the load resistance. It is evident that, when the load resistance Z increases, the voltage rises sufficiently to keep the power constant. When the load is in a state close to short circuit and idle running, the power naturally drops. The voltage in the case of a short circuit also decreases, and in idle running, it increases, but not as much as with the conventional circuits of autonomous inverters [7,19–22].

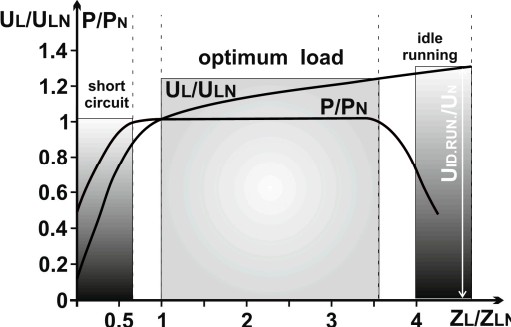

**Figure 10.** Load characteristic of AI with ED.

These results have been used to plot the load characteristic of AI with ED in case of battery charging. Figure 11 presents either of the studied modes of battery charge at two charging power values (current values) until the charge level SOC = 90%, when the process ends. The initial points in both charging scenarios start from SOC = 10% and SOC = 50%. At the beginning of the process of charging (0–500 s), the AI with ED maintains constant power and gradually increases in current value from $0.8I_{DC}$ to reach $I_{DC}$. For the battery LiFePO4, the charging voltage can be calculated by following relationship:

$$U_{BAT\,f} = U_{BAT\,90\%} - I_{DC}R_{BAT} = 310 + 0.9(374 - 310) - I_{DC}R_{BAT} \tag{30}$$

where $U_{BATf}$ is battery voltage in the end of charging process and $U_{BAT\,90\%}$ is battery voltage at SOC = 90%.

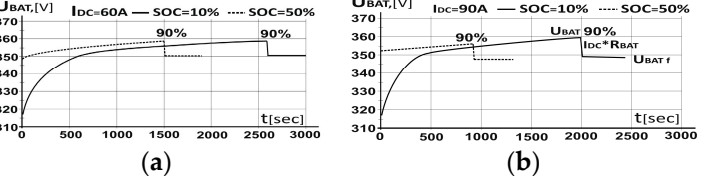

**Figure 11.** Charging scenario with power to the battery: (**a**) 21 kW ($I_{DC}$ = 60 A) and SOC 10 and 50% and (**b**) 31 kW ($I_{DC}$ = 90 A) and SOC 10 and 50%.

From the implemented tests, it can be concluded that the charging source works most efficiently while maintaining maximum charging current or/and output power. Depending on the type of battery used, there is a maximum value of SOC, to which the charging process can be carried out—typically, SOC = 80 ÷ 90%.

The time required for charging the battery depends on its SOC and the power of the charge station. As the transmitting coils are operating in pulse mode, they can be overloaded in order to

achieve shorter charging zone. At the other hand on highways there could be special lane for electric vehicles—by driving in it they can add a certain charge to their batteries and proceed to the nearest charge station. The expressions obtained from the circuit analysis and tests results allow to plot charging characteristics for different batteries. They define the area in which AI with ED can support constant output current and/or power.

It should also be noted that changing the dosing capacitor value $C_{d1}$—$C_{d3}$ leads to a proportional change of the output power in accordance with Equation (25). Furthermore, operating zero current switch (ZCS) mode of the switches is maintained (Figure 12).

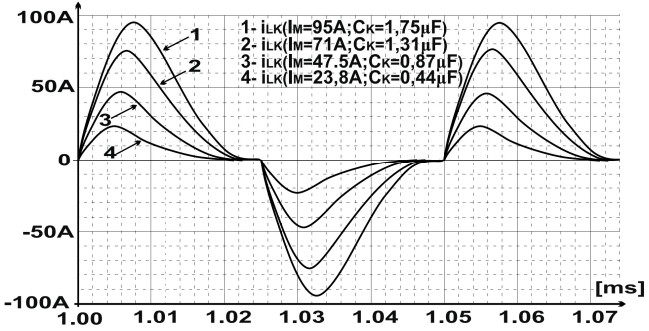

**Figure 12.** The current through AC diagonal of AI with ED at different dosing capacitor: 1–100% $P_{RATED}$; 2–75% $P_{RATED}$; 3–50% $P_{RATED}$; 4–25% $P_{RATED}$.

The research conclusion is that changing the power value by electronic switches might be acceptable as one of the directions to be followed in creating a special HF power supply for EV charging stations.

By changing the values $E$, $C_d$, or $f$, it is possible to regulate the output voltage or to fix an assigned level for it.

$$U_{LOAD} = \sqrt{C_d}\left[E\sqrt{4fR_L}\right]/cos\varphi_L \tag{31}$$

Equation (31) suggests that the output voltage can be maintained at a constant if the $C_d$ capacitor value is varied when the load and/or input DC voltage are changed.

Another very important advantage about flexibility of the developed charging station is that load matching can be controlled with a suitable selection of primary matching capacitor value (15), by electronic switches, during the charging process. The test results are summarized in Figure 13. They represent the electrical parameters $L_1$, $L_2$, $M$, $k$ as a function of the distance between the transmitting and receiving coils (50–200 mm) and horizontal misalignment (0/0–200/200 mm). The condition is to maintain resonance at the HF inverter output in the whole range of possible coil positions. In accordance with the required capacitor values, matching is proposed to be affected with three capacitors—one main capacitor with capacity of 1.1 µF, and two additional ones with capacities of 0.1 µF and 0.2 µF. Figure 13 shows the algorithm for selection of the right capacitor values using appropriate shading. The accuracy and range of this method of dynamic matching depends on the number of electronic switches and capacitors values.

It is important to note that, with different capacitor values, the transistors switch on and off at zero current, and the transient processes do not generate transistor current and voltage overload. Figure 14 shows the current through transistors at the variation of matching capacitors from 25% $C_1$ to 100% $C_1$ and from 100% $C_1$ to 25% $C_1$. It should be noted in both cases that the transient processes finish for one or two periods, ensuring the transistors' safe operating conditions.

| Coil dimensions 700x800x60mm; w₁/w₂=7/7 | | | | | | |
|---|---|---|---|---|---|---|
| C₁ → | | | | | | |
| **dX/dY** | **Z[mm]** | **50** | **100** | **150** | **200** | **250** |
| 0/0 | k | 0.52 | 0.46 | 0.32 | 0.18 | 0.11 |
| | L1[µH] | 61.57 | 61.39 | 61.0 | 59.22 | 58.67 |
| | L2[µH] | 61.73 | 61.30 | 61.08 | 59.99 | 58.97 |
| | M[µH] | 32.06 | 28.22 | 19.53 | 10.73 | 6.47 |
| | C1[µF] | 1.407 | 1.312 | 1.156 | 1.092 | 1.088 |
| | C2[µF] | 1.027 | 1.034 | 1.038 | 1.056 | 1.075 |
| 50/50 | k | 0.498 | 0.394 | 0.29 | 0.15 | 0.09 |
| | L1[µH] | 61.5 | 59.46 | 58.96 | 58.21 | 58.0 |
| | L2[µH] | 61.68 | 60.59 | 59.89 | 59.32 | 58.22 |
| | M[µH] | 30.67 | 23.65 | 17.23 | 8.81 | 5.25 |
| | C1[µF] | 1.367 | 1.239 | 1.155 | 1.093 | 1.089 |
| | C2[µF] | 1.028 | 1.046 | 1.058 | 1.068 | 1.079 |
| 100/100 | k | 0.35 | 0.257 | 0.225 | 0.11 | 0.08 |
| | L1[µH] | 60.46 | 58.26 | 58.04 | 57.38 | 57.02 |
| | L2[µH] | 60.98 | 58.7 | 58.11 | 57.87 | 57.55 |
| | M[µH] | 21.25 | 14.79 | 13.06 | 6.33 | 4.58 |
| | C1[µF] | 1.184 | 1.156 | 1.150 | 1.109 | 1.108 |
| | C2[µF] | 1.039 | 1.079 | 1.09 | 1.095 | 1.101 |
| 150/150 | k | 0.25 | 0.176 | 0.15 | 0.09 | 0.07 |
| | L1[µH] | 59.06 | 57.94 | 57.23 | 56.00 | 55.41 |
| | L2[µH] | 59.66 | 58.07 | 57.69 | 57.00 | 56.9 |
| | M[µH] | 14.84 | 10.21 | 8.62 | 5.08 | 3.93 |
| | C1[µF] | 1.133 | 1.127 | 1.124 | 1.121 | 1.119 |
| | C2[µF] | 1.062 | 1.091 | 1.098 | 1.112 | 1.114 |
| 200/200 | k | 0.066 | 0.032 | 0.025 | 0.02 | 0.01 |
| | L1[µH] | 58.93 | 57.97 | 57.00 | 55.92 | 55.03 |
| | L2[µH] | 59.00 | 58.9 | 59.00 | 59.0 | 59.0 |
| | M[µH] | 3.9 | 1.87 | 1.45 | 1.15 | 0.57 |
| | C1[µF] | 1.079 | 1.077 | 1.075 | 1.074 | 1.074 |
| | C2[µF] | 1.074 | 1.076 | 1.074 | 1.074 | 1.074 |

**Figure 13.** IPT equivalent parameters and matching scenario vs. horizontal and vertical misalignment.

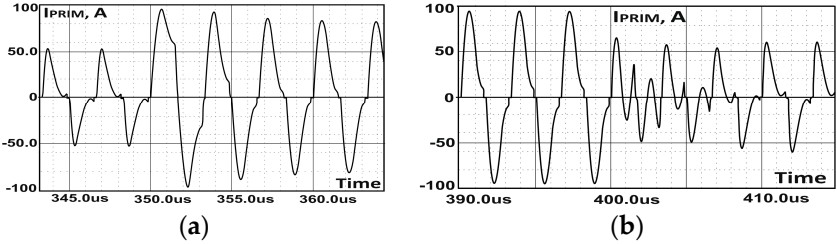

**Figure 14.** Experimental waveforms of the primary coil current at the variation of matching capacitor: (**a**)-from 25% C₁ to 100% C₁ and (**b**)-from 100% C₁ to 25% C₁.

The optimal matching at different positioning between the two coils is related to maximum efficiency, in accordance with Figure 13. Therefore, the range of operational air gap and horizontal misalignment of the IPT system must be selected very carefully. Figure 15 presents the dependence of efficiency as a function of horizontal misalignment between the coils and a vertical distance between them of 100 mm. These measurements are valid for static and dynamic charge, as the configuration of the contactless module remains unchanged. In order to assess the losses, the magnitudes of the currents and voltages and the phase angle between them are measured, as are the input and output of the HF inverter and the IPT module. In addition to this, the IPT module has been tested in short circuit and with no load in order to calculate the parameters of its equivalent circuit. The abovementioned measurements were performed in order to precisely define the equivalent parameters and the losses in each part of the wireless charge station.

The total max efficiency of the charging station from the mains to the battery at zero misalignment is 90–92%. This efficiency is obtained by the charging station modules in the following way: (a) HF

inverter 97–98%; (b) IPT module 94–95%, primary 96–97% (copper app. 98%, ferrite core app. 98%), secondary 97–98% (copper app. 99%, ferrite core app. 98%); (c) output rectifier 98–99%.

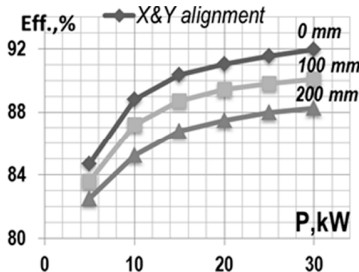

**Figure 15.** IPT efficiency vs output power and X&Y misalignment with 100 mm vertical air gap and power of 5–30 kW.

## 6. Conclusions

The feasibility and advantages of a wireless power transfer charging station using dynamic matching and power supply with energy dosing have been introduced. Based on the analysis carried out, an IPT with compound ferrite cores having a 50–200 mm air gap was designed, which transfered the required electrical power (35 kW) to the battery. The equivalent circuit analysis and characteristics of the IPT system were presented. It has been shown that the magnetic coupling is an important parameter for IPT systems, as it determines the maximum transferred power and can have a great impact on system efficiency. Experiments were carried out to test the ability of the system with different air gaps and misalignments. The results showed that the system can transfer the specified electrical power with about 82–92% efficiency and that the IPT exhibited a high degree of stability under a misaligned (x-y-z) condition.

As an HF power supply, an AI with ED was designed and implemented to verify the validity of the developed operating mode of energy dosing and control algorithm. The obtained expressions formulate the law for keeping the output power constant when the IPT and battery parameters are changed. It has been confirmed in energy dosing mode algorithm that the charging converter works most efficiently and maintains maximum charging current and/or output power.

It was analyzed, tested, and concluded that IPT matching plays an important role when maximizing the transferred power and efficiency. For this reason, electronic switches with series capacitors were used, and an analysis of the dynamic matching was performed. A manner of changing capacitors values with three steps is defined for different scenarios, which are the result of specified x-y-z deviations (50–200 mm in vertical and ±200 mm in horizontal direction).

The obtained results and the drawn conclusions show that the proposed AI with ED and dynamic matching can be used as charging power supply sources characterized by flexibility and the possibility of working with a wide range of IPT and battery parameters.

**Author Contributions:** N.M., and N.H. were involved in the full process of producing this paper including conceptualization, methodology, modelling, validation, visualization, and preparing the manuscript.

**Funding:** The carried-out research is realized in the frames of the project "Model based design of power electronic devices with guaranteed parameters", ДН07/06/15.12.2016, Bulgarian National Scientific Fund.

**Conflicts of Interest:** The authors declare no conflict of interest.

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
