# Peer review of "Flexibility of Wireless Power Transfer Charging Station Using Dynamic Matching and Power Supply with Energy Dosing"

_applsci, doi:10.3390/app9224767_

Round 1

Reviewer 1 Report

Review of the manuscript entitled:

Flexibility of Wireless Power Transfer Charging Station Using Dynamic Matching and Power Supply with Energy Dosing

Manuscript discusses dynamic matching of wireless charging station using inductive power transfer module. The problem was analyzed in detail. The analysis outcomes about the relative values of inductances of equivalent circuit are presented. The prototype of the charging station for transmission of 30-35 kW power at air gap 50-200 mm was designed and carried out. Results of numerical simulations and experiments are presented.

The issue discussed is interesting and current. The work done is extensive. The article presents: overview of current existing solutions, theoretical analysis of the problem, design of the wireless charging station, modeling of the analyzed system with the equivalent circuit, station analysis by means of the equivalent circuit and the conclusion of findings, realization of the prototype of the charging station, testing of the charging station and analysis of the obtained results. The results and findings obtained are useful for electrical engineers and experts dealing with the subject.

I suggest accepting the manuscript for publication.

Smaller adjustments will make the contribution even better, so I suggest that the author consider my suggestions:

- Fig. 1 on page 2 is very simplified, as it is, it does not provide additional information (which would not be presented in the text),

- page 3, line 83: k is not written italic, MTX-RX is not written as in equation,

- equation 1 on page 3: Is the equation written correctly – is in the integral flux or linkage?

- in all subsequent equations the author may omit the dot to indicate multiplication,

- equation 4 on page 4: It is not clear whether this equation is the authors' finding or the equation is already known,

- sometimes incorrect fonts are used in the parameter label text, e.g. k0 on page 5 in line 144, f on page 7 in line 198

Author Response

Answers to reviewers

First of all we would like to thank you for the thorough review of our paper (applsci-627740) and the useful remarks to improve it.

Reviewer 1

Comments to the Authors

Smaller adjustments will make the contribution even better, so I suggest that the author consider my suggestions:

- Fig. 1 on page 2 is very simplified, as it is, it does not provide additional information (which would not be presented in the text),

- page 3, line 83: k is not written italic, MTX-RX is not written as in equation,

- equation 1 on page 3: Is the equation written correctly – is in the integral flux or linkage?

- in all subsequent equations the author may omit the dot to indicate multiplication,

- equation 4 on page 4: It is not clear whether this equation is the authors' finding or the equation is already known,

- sometimes incorrect fonts are used in the parameter label text, e.g. k0 on page 5 in line 144, f on page 7 in line 198

To Reviewer 1:

Thank you for your review and the expressed opinion, that the paper is suitable for publication in Applied Sciences.

Figure 1 is replaced by a more appropriate one; Concerning equation 4 - it was obtained from previous studies by the authors and we have no claim for this manuscript to be new. All your other comments are reflected in the new version of the manuscript.

Again thank you all for the exact review.

Best regards,

Nikolay Hinov, Nikolay Madzharov

Reviewer 2 Report

Paper 627740

The authors present useful and practical work, concerning the development of a wireless power transfer charging stations for electric vehicles. The problem has been worked-out for some recent years, but the authors propose quite novel approach to it. The paper is based on the analytical analysis of the problem.  

Paper is well prepared and is almost in accordance with Applied Science journal template, but some minor remarks should be taken into consideration, namely:

Line 22 – there should be a space between the value and its unit – namely “200 mm”, NOT “200mm” as it is in this line. Please check the whole text and make a correction – e.g. also in lines: 228, 229, 264, 265 etc. Line 32 and 33 – please use square brackets to refer to references number [1] and [2] – there is: (1) and (2)  Line34 – there should be a space between reference number 8 and 9. Figure 1 – please insert an empty line before the figure and after the caption to make it more transparent. This same for other figures. Line 83, 96 and others – the coefficient “k” should be written in italic mode like other coefficients are. Please correct it in the whole text – e.g. line 278. After line 101, Formula (3) – the multiplication symbol as dot/point should be in the centre of the line – please check and correct it in other formulas and in the whole text. Line 264 and 265 – there should be NO space between 50 and % - should be “50%” – check it in the whole text.    Line 270 – please explain the meaning of “ZCS” abbreviation. Line 306 – is “Table 2” – should be “Table 3”!!!! Line 306 – there is “Horizontal”, should be “horizontal”. Line 319 – is “On Fig. 14 …”, should be “In Fig 14 …”. References – shouldn't the names of authors of all reference positions be written with lower-case  letters? Line 394 - there is only one quotation mark. Line 397 - is "VOL", should be "vol".

Author Response

Answers to reviewers

First of all we would like to thank you for the thorough review of our paper (applsci-627740) and the useful remarks to improve it.

Reviewer 2

Comments to the Authors

The authors present useful and practical work, concerning the development of a wireless power transfer charging stations for electric vehicles. The problem has been worked-out for some recent years, but the authors propose quite novel approach to it. The paper is based on the analytical analysis of the problem.

Paper is well prepared and is almost in accordance with Applied Science journal template, but some minor remarks should be taken into consideration, namely:

Line 22 – there should be a space between the value and its unit – namely “200 mm”, NOT “200mm” as it is in this line. Please check the whole text and make a correction – e.g. also in lines: 228, 229, 264, 265 etc.

Line 32 and 33 – please use square brackets to refer to references number [1] and [2] – there is: (1) and (2)

Line34 – there should be a space between reference number 8 and 9.

Figure 1 – please insert an empty line before the figure and after the caption to make it more transparent. This same for other figures.

Line 83, 96 and others – the coefficient “k” should be written in italic mode like other coefficients are. Please correct it in the whole text – e.g. line 278.

After line 101, Formula (3) – the multiplication symbol as dot/point should be in the centre of the line – please check and correct it in other formulas and in the whole text.

Line 264 and 265 – there should be NO space between 50 and % - should be “50%” – check it in the whole text.  

Line 270 – please explain the meaning of “ZCS” abbreviation.

Line 306 – is “Table 2” – should be “Table 3”!!!!

Line 306 – there is “Horizontal”, should be “horizontal”.

Line 319 – is “On Fig. 14 …”, should be “In Fig 14 …”.

References – shouldn't the names of authors of all reference positions be written with lower-case letters?

Line 394 - there is only one quotation mark. Line 397 - is "VOL", should be "vol".

To Reviewer 2:

Thank you for your review of our paper (applsci-627740) and the valuable recommendations.

The all recommendations made are reflected in the text.

Again thank you all for the exact review.

Best regards,

Nikolay Hinov, Nikolay Madzharov

Reviewer 3 Report

This paper presents the design and tests of a fast inductive charging applications for electric vehicles based on an autonomous inverter with energy dosing. The paper analyzes two scenarios regarding static and dynamic charging and concludes that the required powered is transferred with 82-92% efficiency.

R1. The authors should present clearer the efficiency of the system for both cases. I suggest a table to easily evaluate the performance of the system in Static and Dynamic mode.

R2. According to Fig. 11 a) for a changing current of 60A the time to charge a battery from 10% SoC to 90% SoC will be 2500 s almost 42 minutes. In a dynamic scenarios considering a 30 km/h velocity the length of the charging system should be 21 km which seems not feasible. Please clarify these findings.

R3. How the efficiencies for the components of the system were established?

The proposed system is presented in a very detailed way. All the equation that are used for simulation are explained and all the assumptions are carefully chosen. After the simulation a prototype is built and experimental results are obtained. I would like to congratulate the authors for this thorough work and wish them to keep it up.

The article has the main key elements of a review paper: abstract, introduction, simulation, results and conclusions.

The paper is based on good number of references – 20, and three of them belong to the authors.

R4. The authors should consider revising the reference list because some of the titles are more the 10 years old. We suggest to include more papers published after 2010.

I would like to suggest to the authors the following reference, which could be of help for them for both current and future research endeavors:

Seritan G., Porumb R., Cepisca C., Grigorescu S.D. - book: Electricity distribution - Intelligent Solutions for Electricity Transmission and Distribution Networks - chapter: Integration of Dispersed Power Generation, Series: Energy Systems, Springer, 2016, ISBN 978-3-662-49434-9, WOS:000387869200015; Porumb R., Seritan G. - book:Green Energy Advances - chapter: Integration of Advanced Technologies for Efficient Operation of Smart Grids, IET Inspec, EBSCO, DOI: 10.5772/intechopen.77501, ISBN: 978-1-78984-200-5

Are not blurred, with good resolution, clear content, uniformly text included, and all are cited in the text.

R5. The references should be cited in order in text. References 1 and 2 should have [] and not ().

Author Response

Answers to reviewers

First of all we would like to thank you for the thorough review of our paper (applsci-627740) and the useful remarks to improve it.

Reviewer 3

Comments to the Authors

This paper presents the design and tests of a fast inductive charging applications for electric vehicles based on an autonomous inverter with energy dosing. The paper analyzes two scenarios regarding static and dynamic charging and concludes that the required powered is transferred with 82-92% efficiency.

R1. The authors should present clearer the efficiency of the system for both cases. I suggest a table to easily evaluate the performance of the system in Static and Dynamic mode.

R2. According to Fig. 11 a) for a changing current of 60A the time to charge a battery from 10% SoC to 90% SoC will be 2500 s almost 42 minutes. In a dynamic scenarios considering a 30 km/h velocity the length of the charging system should be 21 km which seems not feasible. Please clarify these findings.

R3. How the efficiencies for the components of the system were established?

The proposed system is presented in a very detailed way. All the equation that are used for simulation are explained and all the assumptions are carefully chosen. After the simulation a prototype is built and experimental results are obtained. I would like to congratulate the authors for this thorough work and wish them to keep it up.

The article has the main key elements of a review paper: abstract, introduction, simulation, results and conclusions.

The paper is based on good number of references – 20, and three of them belong to the authors.

R4. The authors should consider revising the reference list because some of the titles are more the 10 years old. We suggest to include more papers published after 2010.

I would like to suggest to the authors the following reference, which could be of help for them for both current and future research endeavors:

Seritan G., Porumb R., Cepisca C., Grigorescu S.D. - book: Electricity distribution - Intelligent Solutions for Electricity Transmission and Distribution Networks - chapter: Integration of Dispersed Power Generation, Series: Energy Systems, Springer, 2016, ISBN 978-3-662-49434-9, WOS:000387869200015;

Porumb R., Seritan G. - book:Green Energy Advances - chapter: Integration of Advanced Technologies for Efficient Operation of Smart Grids, IET Inspec, EBSCO, DOI: 10.5772/intechopen.77501, ISBN: 978-1-78984-200-5

Are not blurred, with good resolution, clear content, uniformly text included, and all are cited in the text.

R5. The references should be cited in order in text. References 1 and 2 should have [] and not ().

To Reviewer 3:

Thank you for your review and recommendation the paper (applsci-627740) to be accepted for publication in Applied Sciences.

The answers to your remarks are as follows:

R1. and R3: The following explanation is added to the text: These measurements are valid for static and dynamic charge, as the configuration of the contactless module remains unchanged. In order to assess the losses, the magnitudes of the currents and voltages and the phase angle between them are measured and the input and output of the HF inverter and the IPT module. In addition to this the IPT module has been tested in short circuit and with no load in order to calculate the parameters of its equivalent circuit. The abovementioned measurements were performed in order to precisely define the equivalent parameters and the losses in each part of the wireless charge station.

R2: The following explanation is added to the text: The time required for charging the battery depends on its SOC and the power of the charge station. As the transmitting coils are operating in pulse mode they can be overloaded in order to achieve shorter charging zone. At the other hand on highways there could be special lane for electric vehicles – by driving in it they can add a certain charge to their batteries and proceed to the nearest charge station.

R4. and R5: We accept the remark and an adjustment has been made.

Again thank you all for the exact review.

Best regards,

Nikolay Hinov, Nikolay Madzharov